# Phytochemistry and Comprehensive Chemical Profiling Study of Flavonoids and Phenolic Acids in the Aerial Parts of *Allium Mongolicum* Regel and Their Intestinal Motility Evaluation

**DOI:** 10.3390/molecules25030577

**Published:** 2020-01-29

**Authors:** Yongzhe Dong, Jingya Ruan, Zhijuan Ding, Wei Zhao, Mimi Hao, Ying Zhang, Hongyu Jiang, Yi Zhang, Tao Wang

**Affiliations:** 1Tianjin Key Laboratory of TCM Chemistry and Analysis, Tianjin University of Traditional Chinese Medicine, 10 Poyanghu Road, West Area, Tuanbo New Town, Jinghai District, 301617 Tianjin, China; dongyongzhe44@hotmail.com (Y.D.); Ruanjy19930919@163.com (J.R.); zhaowei126123@126.com (W.Z.); 2Institute of TCM, Tianjin University of Traditional Chinese Medicine, 10 Poyanghu Road, West Area, Tuanbo New Town, Jinghai District, 301617 Tianjin, China; 15222792071@163.com (Z.D.); haomimi126@126.com (M.H.); zyingtzy@163.com (Y.Z.); jhy15731602454@163.com (H.J.)

**Keywords:** *Allium mongolicum* Regel, flavonoids, phenolic acids, mouse isolated intestine tissue, qualitative analysis

## Abstract

To clarify whether flavonoids and phenols in *Allium mongolicum* Regel have the effect of improving gastrointestinal function and analyze its quality, this study was designed to isolate and identify them from the aerial parts of *A. mongolicum* by using various chromatographic and spectrophotometric methods, a bioassay on motility of mouse isolated intestine tissue, as well as qualitative analysis using liquid chromatography/mass spectrometry (LC-MS) analysis. As a result, 31 flavonoids and phenolic acids were obtained and identified, including six new flavonoid glycosides, mongoflavonosides A_1_ (**1**), A_2_ (**2**), A_3_ (**3**), A_4_ (**4**), B_1_ (**5**), B_2_ (**6**), and four new phenolic acid glycosides, mongophenosides A_1_ (**7**), A_2_ (**8**), A_3_ (**9**), B (**10**). Among them, eleven flavonoids and three phenolic acids showed significant increase in the height of mouse small intestinal muscle. It was a first systematic bioactive constituents’ study for *A. mongolicum* on gastrointestinal tract. Furthermore, according to the retention time (*t*_R_) and the exact mass-to-charge ratio (*m/z*), thirty-one compounds were unambiguously identified by comparing to the standard references by using LC-MS. Then, on the basis of generalized rules of MS/MS fragmentation pattern, chromatographic behaviors, as well as biosynthetic laws of the 31 isolates, five flavonoid glycosides and one phenolic acid glycoside were tentatively speculated. On the basis of the study, a fast analysis method for flavonoids and phenolic acids in *A. mongolicum* was established.

## 1. Introduction

Flavonoids and phenolic acids are secondary metabolites found in most *Allium* vegetables, such as onion (*Allium cepa* L.) [1], scallion (*Allium fistulosum* L.) [2], garlic (*Allium sativum* L.) [3], fruits, and traditional medicine. They also exert multiple biological properties, such as antitumor [4,5] antioxidant [6,7], anti-inflammation and gastrointestinal motility effect improvement [8,9,10,11], which make them show high correlations in the inhibition or management of many chronic diseases, such as cardiovascular and cerebrovascular diseases, diabetes, cancer, digestive system diseases, and so on.

As a traditional Mongolian medicinal herb, *Allium mongolicum* Regel (Liliaceae family) is mainly found in the high altitude desert of the Inner Mongolia Autonomous Region, Ningxia Hui Autonomous Region, Qinghai, Gansu, and Shanxi provinces [12], and has various special properties, such as stimulating the appetite, lowering blood pressure, hypolipidemic, replenishing the kidneys, and acting as an aphrodisiac [7]. Though it has been used to stimulate the appetite, studies were mainly focused on animal cultivation and improvement on meat products [12,13,14] up until now. Experiments related to the gastrointestinal tract have not been reported for the plant. On the other hand, pharmacological investigations showed the biological activities of *A. mongolicum*, including anti-inflammatory, antimicrobial, and antioxidant, which were attributed to the presence of different phytochemical groups like flavonoids and phenols [7,15,16]. However, it is still limited to the activity of total flavonoids and phenolic acids. Until now, the phytochemistry investigation of flavonoids is only reported by Dr. Zhao [16] and our laboratory [17,18]—there is no report for phenolic acids investigation. Moreover, there is no comprehensive chemical profiling study for the plant.

The present study was designed to isolate and identify flavonoids and phenolic acids from the aerial parts of *A. mongolicum* followed by bioactivity study on the motility of mouse isolated intestine tissue and qualitative analysis using liquid chromatography/mass spectrometry (LC-MS) analysis.

## 2. Results and Discussion

The fresh, aerial parts of *A. mongolicum* (17.8 kg) was successively heated under reflux with 95% EtOH for 3 h and 50% EtOH for 2 h one time each to obtain dry extract of *A. mongolicum* aerial parts (AM, 515.0 g). Then 470.1 g of it was partitioned with EtOAc/H_2_O (1:1, 8L/8L) to yield EtOAc layer dry extract (AME, 64.9 g) and H_2_O layer dry extract (AMH, 381.0 g).

Then, AM, AMH, and AME were tested for frequency and height by using a tissue perfusion method. As results, AM and AMH showed significant increase in the contraction amplitude of mouse small intestinal muscle at 200 μg/mL (relative height for AM: 137.4 ± 11.8%* and AMH: 121.8 ± 1.0%**, respectively), but had no significant effect on frequency (relative frequency for AM: 95.2 ± 2.8% and AMH: 100.1 ± 1.9%, respectively). While AME displayed no significant effect on both of them (relative height: 127.9 ± 20.8%; relative frequency: 100.0 ± 9.18%).

Therefore, AMH was further fractionated by column chromatography (CC) and purified by HPLC to afford six new flavonoid glycosides, named as mongoflavonosides A_1_–A_4_ (**1**–**4**), B_1_ (**5**), B_2_ (**6**), four new phenolic acid glycosides, named as mongophenosides A_1_–A_3_ (**7**–**9**), B (**10**) (Figure 1), along with known compounds, kaempferol-3,7,4′-tri-*O*-β-glucoside (**18**) [19], quercetin-3-*O*-β-d-rutinoside-7-*O*-β-d-glucuronide (**25**) [20], quercetin-3,7,4′-tri-*O*-glucoside (**26**) [21], isorhamnetin 3-*O*-β-d-glucopyranoside (**27**) [22], as well as three phenols, *trans*-*p*-hydroxycinnamate sophorose (**28**) [23], tuberonoid A (**29**) [24], trans-caffeic acid (**30**) [25], and benzyl-*O*-β-d-glucopyranoside (**31**) [26] (Figure 2). Among the known isolates, **28** and **30** were obtained from the *Allium* genus for the first time; **18**, **25**–**27**, **29**, and **31** were firstly found from the plant.

Furthermore, the improved effects on the motility of the mouse isolated intestine tissue of the above-mentioned compounds as well as our previously reported flavonoids, kaempferol-3-*O*-β-d-glucopyranoside (**11**), kaempferol-3-*O*-β-d-glucopyranosyl(1→4)-β-d-glucopyranoside (**12**), kaempferol-3-*O*-β-d-rutinoside (**13**), kaemperol-3-*O*-β-d-glucopyranosyl(1→4)[α-l-rhamanopyranosyl(1→6)]-β-d-glucopyranoside (**14**), kaempferol-3-*O*-rutinoside-7-*O*-glucuronide (**15**), kaempferol-3-rutinoside-4′-glucopyranoside (**16**), kaempferol-3-*O*-gentiobioside-4′-*O*-glucopyranoside (**17**) [17], isoquercetin (**19**), quercetin-3-*O*-(6′′-*O*-acetyl)-β-d-glucopyranoside (**20**), quercetin-3-*O*-β-d-glucopyranosyl(1→4)-β-d-glucopyranoside (**21**), rutin (**22**), quercetin-3,4′-di-*O*-β-d-glucopyranoside (**23**), quercetin 3-*O*-(6′′-*O*-α-l-rhamnopyranosyl)-β-d-glucopyranoside-7-*O*-β-d-glucopyranoside (**24**) [18] (Figure 2) were reported here. Then, qualitative analysis for the aerial parts of *A. mongolicum* by using LC-MS spectrometry technology was developed.

### 2.1. Identification of Compounds

Mongoflavonoside A_1_ (**1**) was isolated as a yellow powder with negative optical rotation ([α]_D_^25^ −54.0, H_2_O). Its molecular formula was deduced to be C_33_H_38_O_22_ by the negative-ion Electron Spray Ionization-Quadrupole-Orbitrap-Mass Spectrometry (ESI-Q-Orbitrap MS) analysis (*m/z* 785.17883 [M − H]^−^, calculated for C_33_H_37_O_22_, 785.17710). The IR spectrum displayed the absorption bands assignable to hydroxyl (3354 cm^−1^), carbonyl (1716 cm^−1^), α,β-unsaturated ketone carbonyl (1652 cm^−1^), aromatic ring (1601, 1507, 1457 cm^−1^), and ether functions (1072 cm^−1^), respectively. The ^1^H and ^13^C-NMR (Table 1) spectra suggested that **1** was a flavonoid glycoside with a kaempferol aglycone [δ 6.48 (1H, br. s, H-6), 6.88 (1H, br. s, H-8), 7.19 (2H, d, *J* = 9.0 Hz, H-3′,5′), 8.16 (2H, d, *J* = 9.0 Hz, H-2′,6′)] and three glycosyl groups [δ 5.05 (1H, d, *J* = 7.0 Hz, H-1′′′′), 5.26 (1H, d, *J* = 6.5 Hz, H-1′′′), 5.50 (1H, d, *J* = 7.0 Hz, H-1′′)]. Acid hydrolysis of **1** with 5% aqueous H_2_SO_4_ solution–1,4-dioxane (1:1, *v/v*) under 110 °C for 2 h to afford d-glucuronic acid and d-glucose, whose absolute configurations were determined by GC-MS analysis of their trimethysilyl thiazolidine derivatives [27]. Meanwhile, correlations were observed between the following proton and carbon pairs in its HSQC-TOCSY spectrum: H-1′′ and C-1′′–C-5′′; δ_H_ 3.35, 3.58 (H_2_-6′′) and C-4′′–C-6′′; H-1′′′ and C-1′′′–C-4′′′; δ_H_ 3.98 (H-5′′′) and C-2′′′–C-5′′′; H-1′′′′ and C-2′′′′–C-4′′′′; δ_H_ 3.50 (H_2_-6′′′′) and C-4′′′′–C-6′′′′. Combining with the correlations displayed in its ^1^H ^1^H COSY and HSQC spectrum, the NMR data of three glycosyls were assigned in detail. Finally, according to the long-range correlations from H-1′′ to C-3; H-1′′′ to C-7; H-1′′′′ to C-4′ showed in its HMBC experiment (Figure 3), the connections between glycosyl groups and aglycone were determined. On the basis of above-mentioned evidence, the structure of mongoflavonoside A_1_ (**1**) was identified.

Mongoflavonoside A_2_ (**2**) is a yellow powder with negative optical rotation ([α]_D_^25^ −26.0, MeOH). Its ESI-Q-Orbitrap MS spectrum showed a peak at *m/z* 771.19971 [M − H]^−^ (calculated for C_33_H_39_O_21_, 771.19783), and its molecular formula was deduced to be C_33_H_40_O_21_. After hydrolyzing with 1 M HCl, the product was analyzed by using HPLC with an optical rotation detector. As a result, d-glucose was detected [28]. The ^1^H and ^13^C-NMR (Table 1) spectra suggested that compound **2** had the same aglycone, kaempferol [δ 6.21 (1H, br. s, H-6), 6.44 (1H, br. s, H-8), 7.17 (2H, d, *J* = 9.0 Hz, H-3′,5′), 8.11 (2H, d, *J* = 9.0 Hz, H-2′,6′)] as that of **1**. In addition, there were three β-d-glucopyranosyl moieties [δ 4.27 (1H, d, *J* = 8.0 Hz, H-1′′′), 5.03 (1H, d, *J* = 7.5 Hz, H-1′′′′′), 5.51 (1H, d, *J* = 8.0 Hz, H-1′′)]. To solve the overlapping problem of three β-d-glucopyranosyl groups, the HSQC-TOCSY experiment was performed. The correlations between C-1′′ and δ_H_ 3.26 (H-2′′), 3.39 (H-4′′), 3.41 (H-3′′), 5.51 (H-1′′); δ_H_ 3.51, 3.63 (H_2_-6′′) and C-4′′–C-6′′; H-1′′′ and C-1′′′–C-4′′′; δ_H_ 3.42, 3.71 (H_2_-6′′′) and C-4′′′–C-6′′′; H-1′′′′′ and C-1′′′′′–C-4′′′′′; δ_H_ 3.50, 3.71 (H_2_-6′′′′′) and C-4′′′′′–C-6′′′′′ were found in it. Moreover, the HMBC displayed the long-range correlations from H-1′′ to C-3; H-1′′′ to C-4′′; H-1′′′′′ to C-4′ (Figure 3). Consequently, the structure of mongoflavonoside A_2_ (**2**) was elucidated to be kaempferol 3-*O*-β-d-glucopyranosyl(1→4)-β-d-glucopyranosyl-4′-*O*-β-d-glucopyranoside.

Mongoflavonoside A_3_ (**3**) exhibited negative optical rotation ([α]_D_^25^ −64.7, H_2_O). Its molecular formula was revealed to be C_39_H_48_O_27_ by negative ESI-Q-Orbitrap MS analysis (*m/z* 947.23242 [M − H]^−^, calculated for C_39_H_47_O_27_, 947.22992). The ^1^H, ^13^C-NMR (Table 1) along with various 2D NMR (^1^H ^1^H COSY, HSQC, HMBC, and HSQC-TOCSY) spectra denoted that **3** had the same moiety, kaempferol 3-*O*-β-d-glucopyranosyl(1→4)-β-d-glucopyranosyl-4′-*O*-β-d-glucopyranosyl [δ 4.26 (1H, d, *J* = 7.0 Hz, H-1′′′), 5.04 (1H, d, *J* = 7.0 Hz, H-1′′′′′), 5.53 (1H, d, *J* = 7.5 Hz, H-1′′), 6.46 (1H, br. s, H-6), 6.85 (1H, br. s, H-8), 7.18 (2H, d, *J* = 8.5 Hz, H-3′,5′), 8.14 (2H, d, *J* = 8.5 Hz, H-2′,6′)] as that of **2**. Meanwhile, one more β-d-glucuropyranosyl [δ 5.12 (1H, d, *J* = 7.0 Hz, H-1′′′′)] appeared in **3**. On the other hand, the proton signals at H-6 and H-8 and the carbon signal at C-7 shifted to the lower field in comparison with those of **2**, which suggested that the β-d-glucuropyranosyl linked with 7-position of kaempferol. It was clarified by the long-range correlation from H-1′′′′ to C-7 (Figure 3). Then, the structure of mongoflavonoside A_3_ (**3**) was determined.

The molecular formula of mongoflavonoside A_4_ (**4**) was measured to be C_39_H_48_O_26_ by negative ESI-Q-Orbitrap MS analysis (*m/z* 931.23785 [M − H]^−^, (calculated for C_39_H_47_O_26_, 931.23501). Its acid hydrolysis product was derived to obtain trimethylsilane thiazolidine derivatives, then the existence of d-glucuronic acid, d-glucose, and l-rhamnose were clarified by GC analysis [27]. Its ^1^H, ^13^C-NMR (Table 1) and 2D NMR spectra indicated that **4** had the same moiety, kaempferol 3-*O*-β-d-glucopyranosyl(1→4)[α-l-rhamanopyranosyl(1→6)]-β-d-glucopyranosyl [δ 4.14 (1H, d, *J* = 7.5 Hz, H-1′′′), 4.40 (1H, br. s, H-1′′′′), 5.23 (1H, d, *J* = 7.5 Hz, H-1′′), 6.37 (1H, br. s, H-6), 6.73 (1H, br. s, H-8), 6.88 (2H, d, *J* = 9.0 Hz, H-3′,5′), 8.00 (2H, d, *J* = 9.0 Hz, H-2′,6′)] as that of kaempferol 3-*O*-β-d-glucopyranosyl(1→4)[α-l-rhamanopyranosyl(1→6)]-β-d-glucopyranoside [29]. In addition, one β-d-glucuropyranosyl [δ_H_ 5.16 (1H, d, *J* = 7.0 Hz, H-1′′′′′); δ_C_ 71.6 (C-4′′′′′), 72.8 (C-2′′′′′), 73.7 (C-5′′′′′), 76.2 (C-3′′′′′), 98.7 (C-1′′′′′), 171.7 (C-6′′′′′)] appeared in **4**. The long-range correlation observation from H-1′′′′′ to C-7 (Figure 3) in its HMBC spectrum suggested the β-d-glucuropyranosyl connected with C-7 of kaempferol 3-*O*-β-d-glucopyranosyl(1→4)[α-l-rhamanopyranosyl(1→6)]-β-d-glucopyranosyl. Then, the structure of mongoflavonoside A_4_ (**4**) was constructed.

Mongoflavonoside B_1_ (**5**) was isolated as a yellow powder and showed negative optical rotation ([α]_D_^25^ −12.0, MeOH). The molecular formula, C_29_H_32_O_18_ of **5** was determined from ESI-Q-Orbitrap MS (*m/z* 667.15228 [M − H]^−^; calculated for C_29_H_31_O_18_, 667.15049) analysis. Its IR spectrum exhibited characteristic absorptions of hydroxyl (3362 cm^−1^), ester carbonyl (1721 cm^−1^), α,β-unsaturated ketone carbonyl (1654 cm^−1^), aromatic ring (1605, 1507, 1448 cm^−1^), and ether functions (1070 cm^−1^). The ^1^H and ^13^C-NMR spectra displayed signals of a quercetin moiety [δ 6.18 (1H, br. s, H-6), 6.39 (1H, br. s, H-8), 6.83 (1H, d, *J* = 8.5 Hz, H-5′), 7.50 (1H, dd, *J* = 2.0, 8.5 Hz, H-6′), 7.51 (1H, d, *J* = 2.0 Hz, H-2′)], two β-d-glucopyranosyl groups [δ 4.21 (1H, d, *J* = 8.0 Hz, H-1′′′), 5.40 (1H, d, *J* = 8.0 Hz, H-1′′)], along with an acetyl [δ_H_ 1.71 (3H, s, 6′′-COC*H*_3_); δ_C_ 19.9 (6′′-CO*C*H_3_), 169.6 (6′′-*C*OCH_3_)]. As shown in Figure 3, the ^1^H ^1^H COSY experiment on **5** indicated the presence of partial structures written in bold lines. Moreover, in the HMBC spectrum, the long-range correlations from H-1′′ to C-3; H-1′′′ to C-4′′; 6′′-COC*H*_3_ to 6′′-*C*OCH_3_; H_2_-6′′ to 6′′-*C*OCH_3_ were observed. Finally, after treating **5** with 1 M HCl, d-glucose was detected from its acid hydrolysis product [28]. Consequently, the structure of **5** was identified, and named as mongoflavonoside B_1_.

The molecular formula, C_33_H_38_O_23_ of **6** was measured on ESI-Q-Orbitrap MS (*m/z* 801.17407 [M − H]^−^, calculated for C_33_H_37_O_23_, 801.17201) analysis. The ^1^H, ^13^C NMR (Table 1) and 2D NMR (^1^H ^1^H COSY, HSQC, HMBC, HSQC-TOCSY) spectra suggested **6** had the same glycosyl moieties with **1**: two β-d-glucopyranosyls [δ 4.88 (1H, d, *J* = 7.0 Hz, H-1′′′′), 5.52 (1H, d, *J* = 7.0 Hz, H-1′′)], and one β-d-glucuropyranosyl [5.16 (1H, d, *J* = 7.0 Hz, H-1′′′)]. Meanwhile, **6** possessed the same aglycone, quercetin [δ 6.45 (1H, br. s, H-6), 6.85 (1H, br. s, H-8), 7.23 (1H, d, *J* = 8.5 Hz, H-5′), 7.64 (1H, d, *J* = 1.5, 8.5 Hz, H-6′), 7.69 (1H, d, *J* = 1.5 Hz, H-2′)] as that of **5**. Finally, the connectivities of glycosyl moieties with aglycone were determined by the correlations from H-1′′ to C-3; H-1′′′ to C-7; H-1′′′′ to C-4′ (Figure 3) showed in its HMBC spectrum.

Mongophenoside A_1_ (**7**) was obtained as a white powder with negative optical rotation ([α]_D_^25^ −21.0, MeOH). ESI-Q-Orbitrap MS of **7** exhibited quasimolecular ion peak at *m/z* 503.14151 [M − H]^−^ (calculated for C_21_H_27_O_14_, 503.13953), and its molecular formula was deduced to be C_21_H_28_O_14_. The IR spectrum of it showed absorption bands ascribable to hydroxyl (3362 cm^−1^), α,β-unsaturated ester carbonyl (1709 cm^−1^), aromatic ring (1601, 1521, 1447 cm^−1^), and ether function (1074 cm^−1^). Acid hydrolysis of **7** liberated d-glucose, which was identified by HPLC analysis [28]. Its ^1^H, ^13^C NMR (Table 2) spectra indicated the existence of one *trans*-caffeoyl [δ_H_ 6.27 (1H, d, *J* = 16.0 Hz, H-8), 6.75 (1H, d, *J* = 7.5 Hz, H-5), 7.01 (1H, br. d, ca. *J* = 8 Hz, H-6), 7.06 (1H, br. s, H-2), 7.55 (1H, d, *J* = 16.0 Hz, H-7); δ_C_ 113.3 (C-8), 146.2 (C-7), 164.9 (C-9)], along with two β-d-glucopyranosyl groups [δ 4.42 (1H, d, *J* = 8.0 Hz, H-1′′), 5.56 (1H, d, *J* = 8.0 Hz, H-1′)]. Meanwhile, the partial structures written in bold lines shown in Figure 4 were determined by proton and proton correlations observed in its ^1^H ^1^H COSY experiment. The planar structure of **5** was finally elucidated according to the long-range correlations from H-1′ to C-9; H-1′′ to C-2′ (Figure 4) found in HMBC experiment, and the structure of **7** was named as mongophenoside A_1_.

Mongophenoside A_2_ (**8**), a white powder, showed negative optical rotation ([α]_D_^25^ −14.5, MeOH). ESI-Q-Orbitrap MS analysis suggested its molecular formula was C_27_H_38_O_19_ (665.19427 [M − H]^−^; calculated for C_27_H_37_O_19_, 665.19236). The ^1^H and ^13^C-NMR (Table 2) spectra indicated **8** possessed the same moiety, trans-caffeic acid-9-*O*-β-d-glucopyranosyl(1→2)-β-d-glucopyranosyl [δ 4.42 (1H, d, *J* = 7.5 Hz, H-1′′), 5.56 (1H, d, *J* = 7.0 Hz, H-1′), 6.27 (1H, d, *J* = 16.0 Hz, H-8), 6.76 (1H, d, *J* = 7.5 Hz, H-5), 7.02 (1H, br. d, ca. *J* = 8 Hz, H-6), 7.06 (1H, br. s, H-2), 7.55 (1H, d, *J* = 16.0 Hz, H-7)] as that of **7**. Except for that, one more β-d-glucopyranosyl [δ 4.17 (1H, d, *J* = 7.5 Hz, H-1′′′)] appeared in **8**. Meanwhile, C-6′ of it was found to significantly shift to lower field (δ_C_ 67.7 for **8**; 60.3 for **7**) comparing with **7**, which suggested C-6′ was substituted by the β-d-glucopyranosyl. In the HMBC spectrum, the long-range correlations from H-1′′′ to C-6′; H-1′′ to C-2′; H-1′ to C-9 (Figure 4) were observed. Moreover, treated **8** with 1 M HCl, d-glucose was yielded [28]. Consequently, the structure of mongophenoside A_2_ (**8**) was elucidated.

The ESI-Q-Orbitrap MS spectrum of mongophenoside A_3_ (**9**) displayed the same molecular formula, C_27_H_38_O_19_ (*m/z* 665.19452 [M − H]^−^; calculated for C_27_H_37_O_19_, 665.19236) as that of **8**. Meanwhile, the ^1^H, ^13^C NMR (Table 2) and 2D NMR (^1^H ^1^H COSY, HSQC, HMBC, HSQC-TOCSY) spectra suggested they had same functional groups as following: trans-caffeic acid aglycone [δ 6.45 (1H, d, *J* = 16.0 Hz, H-8), 7.19 (1H, br. s, H-2), 7.12 (2H, m, H-5 and H-6), 7.61 (1H, d, *J* = 16.0 Hz, H-7)] and three β-d-glucopyranosyl groups [δ 4.43 (1H, d, *J* = 7.5 Hz, H-1′′′), 4.80 (1H, d, *J* = 7.5 Hz, H-1′), 5.57 (1H, d, *J* = 7.0 Hz, H-1′′)]. Finally, the connectivities of the above-mentioned groups were clarified by the long-range correlations from H-1′ to C-4; H-1′′ to C-9; H-1′′′ to C-2′′ (Figure 4), as shown in its HMBC experiment.

Mongophenoside B (**10**) was obtained as a white powder with positive optical rotation ([α]_D_^25^ +8.0, MeOH). Its ESI-Q-Orbitrap MS spectrum showed the negative ion peak at *m/z* 517.15668 [M − H]^−^ (calculated for C_22_H_29_O_14_, 517.15518), which indicated the molecular formula of it was C_22_H_30_O_14_. Acid hydrolysis **10** with 1 M HCl, d-glucose was liberated [28]. The ^1^H, ^13^C NMR (Table 2) and 2D NMR spectra of it suggested the existence of one trans-feruloyl [δ 6.34 (1H, d, *J* = 16.0 Hz, H-8), 6.80 (1H, d, *J* = 8.0 Hz, H-5), 7.06 (1H, br. d, ca. *J* = 8 Hz, H-6), 7.17 (1H, br. s, H-2), 7.60 (1H, d, *J* = 16.0 Hz, H-7), 3.88 (3H, s, 3-OC*H*_3_)], one β-d-glucopyranosyl [δ 4.48 (1H, d, *J* = 8.0 Hz, H-1′)], together with one *α*-d-glucopyranosyl [δ 5.10 (1H, *J* = 3.5 Hz, H-1′′)]. Moreover, the long-range correlations from H-1′ to C-9; H-1′′ to C-2′ were observed in its HMBC experiment. On the basis of above-mentioned evidence, the structure of mongophenoside B (**10**) was identified as trans-ferulic acid-9-*O*-α-d-glucopyranosyl(1→2)-β-d-glucopyranoside.

### 2.2. Inhibitory Effects of Obtained Compounds **1**–**31** on the Motility of Mouse Isolated Intestine Tissue

Moreover, the obtained constituents of **1**–**31** were tested for frequency and height by using a tissue perfusion method [30]. Through tissue perfusion experiments, it was found that all compounds displayed no effect on isolated intestinal tissue contraction frequency (Table 3). While almost all isolates exhibited the tendency of increasing the contraction amplitude of mouse small intestinal muscle though only flavonoids **3**, **4**, **11**–**15**, **21**–**23**, and **26**, as well as phenolic acids **7**, **29**, and **30** showed significant difference comparing with normal group.

### 2.3. Qualitative Analysis

As an important edible medicinal plant for Mongolian people, *A. mongolicum* has made a great contribution to the development of the local economy, yet there is a lack of analysis of its quality until now.

Our systematic phytochemistry isolation results indicated the main constituents of AM were flavonoids and phenolic acids. The aglycones in the plant mainly included quercetin, kaempferol, as well as isorhamnetin for flavonoid glycosides; while coumaric acid, caffeic acid, and ferulic acid for phenolic acid glycosides. The sugars consisted of β-d-glucopyranoside (Glc), α-d-glucopyranoside (α-Glc), β-d-glucuronic acid (Glu), and α-l-rhamnopyranoside (Rha). While α-Glc was only found in phenolic acid glycosides, Glu and Rha substituted only for flavonoid glycosides.

As for flavonoids, 3-, 7-, and 4′-OH of quercetin, kaempferol, and isorhamnetin were easily substituted by various glycosyls to format *O*-glycosides. Among them, 7- and 4′-OH was substituted by monosaccharose such as Glc and Glu, while Glu was found to only link with their 7-position. Meanwhile, 3-OH was with a high degree of glycosylation, having one to three sugar moieties, and all of the glycosyl groups directly linked to flavonoid was Glc group, then its 2-, 4-, or 6-position was substituted by another Glc continuously; moreover, its 6-position could also be replaced by Rha [to form rutinosyl (Rut)] or acetyl group (Figure 5).

On the other hand, the carboxyl of obtained phenolic acids from AM was easily substituted by sugar moiety such as Glc(1→2)Glc–, α-Glc(1→2)Glc–, or Glc(1→6)Glc(1→2)Glc– on their 9-position, while 4-OH of them was only substituted by monosaccharose, Glc (Figure 6).

Herein, on the basis of above-mentioned phytochemistry study, a fast analysis method for flavonoids and phenolic acids in AM was established by LC-MS on an ESI-Q-Orbitrap MS in negative ion mode (Figure 7). According to the chromatographic retention time (*t*_R_) and the exact mass-to-charge ratio (*m/z*), 31 compounds (**1**–**31**) were unambiguously identified by comparing to the standard references. Meanwhile, the rules of the MS/MS fragmentation pattern and chromatographic elution order have been generalized. Then, five flavonoid glycosides (**32**–**36**) and one phenolic acid glycoside (**37**) were tentatively speculated. Among them, **36** was a potential new compound (Appendix A, Figure 8).

#### 2.3.1. Structural Elucidation of Flavonoids

Peaks 3′–6′, 9′, 10′, 13′, 16′, 21′, 23′–27′ and 29′–37′ were identified by comparison with reference standards (Appendix A, Figure 7).

Figure 9 and Appendix A showed the MS/MS fragmentation pattern of flavonoid glycosides with kaempferol and quercetin aglycones, which suggested both of two kinds of flavonoid glycosides could be ionized to generate heterolytic cleavage with fragments ion peak at *m/z* 285.03936 (Y_K0_^−^) for kaempferol and *m/z* 301.03428 (Y_Q0_^−^) for quercetin, as well as hemolytic cleavage with fragments ion peak at *m/z* 284.03154 [Y_K0_^−^ − H]^−^ for kaempferol and *m/z* 300.02645 [Y_Q0_^−^ − H]^−^ for quercetin, respectively. Then, kaempferol aglycone could be further cleavage to generate fragment ion peaks at *m/z* 255.02880, 179.02371, and 151.00259. The fragment ion peaks at *m/z* 271.02371, 255.02880, 243.02880, 179.02371, as well as 151.00259, were yielded from quercetin by a series of reactions including decarbonylation, dehydrogenation, retro Diels–Alder reaction, and the reaction to remove the B ring. The above-mentioned characteristic fragment ions could be used to distinguish the type of aglycone.

Meanwhile, when the 4′-position of flavanol aglycone was glycosylated to format *O*-glycoside, the debris ions peaks ([Y_K0_^−^2H]^−^) at *m/z* 283.02371 for kaempferol and *m/z* 299.01863 ([Y_Q0_^−^2H]^−^) for quercetin glycosides were stronger than those of *m/z* 284.03154 and 300.02645, respectively (Appendix A). Therefore, their ionic strength could be used to quickly determine whether the C-4′ position of the aglycone was replaced by sugar.

Peaks 9′, 22′, 23′, and 27′ were obtained by extracting ion of *m/z* 771.19783 from the total ion chromatogram of AM (Figure 10), among them, 9′, 23′, and 27′ were clarified to be kaempferol-3,7,4′-tri-*O*-β-glucoside (**18**), kaempferol-3-*O*-gentiobioside-4′-*O*-glucopyranoside (**17**), and mongoflavonoside A_2_ (**2**) by comparing with reference standards. Then, according to the above-mentioned biosynthetic pathway of substituted sugar, peak 22′ was tentatively presumed to be kaempferol-3-*O*-β-d-glucopyranosyl(1→2)-*O*-β-d-glucopyranosyl-4′-*O*-β-d-glucopyranoside (**36**), which was one new compound.

Moreover, during the comparison of the chromatographic retention behavior of peaks 9′, 22′, 23′, and 27′, we discovered the effect of sugar substitution position on *t*_R_ was 3,7,4′-tri-*O*-Glc < 3-*O*-Glc(1→2)-Glc-4′-*O*-Glc < 3-*O*-Glc(1→6)-Glc-4′-*O*-Glc < 3-*O*-Glc(1→4)-Glc-4′-*O*-Glc.

The molecular formula of peaks 3′ (*m/z* 801.17407), 8′ (*m/z* 801.17462), 11′ (*m/z* 801.17389), and 14′ (*m/z* 801.17200) were all C_33_H_38_O_23_ (Figure 11). Peak 3′ was unambiguously identified as mongoflavonoside B_2_ (**6**) by comparison with reference standard. According to the MS/MS fragment ion peaks at *m/z* 301.03428, 300.02645, 299.01863, 271.02371, 255.02880, and 151.00259, peaks 8′, 11′, and 14′ were deduced to be with quercetin aglycone. On the other hand, the fragment ion peaks at *m/z* 625.13993 [M − H − 176]^−^, 301.03428 [M − H − 176 − 162 − 162]^−^ suggested the presences of one β-d-Glu and two β-d-Glc in them. Since the strength of fragment ion peak at *m/z* 299.01863 was weaker than that of *m/z* 300.02645, we could propose that 4′-OH of quercetin was not be glycosidated. According to the above-mentioned biosynthetic pathway of substituted sugar and effect of sugar substitution position on *t*_R_, peaks 8′, 11′, and 14′ were tentatively presumed to be quercetin-3-*O*-β-d-glucopyranosyl(1→2)-β-d-glucopyranosyl-7-*O*-β-d-glucuronide (**32**), quercetin-3-*O*-β-d-glucopyranosyl(1→6)-β-d-glucopyranosyl-7-*O*-β-d-glucuronide (**33**), and quercetin-3-*O*-β-d-glucopyranosyl(1→4)-β-d-glucopyranosyl-7-*O*-β-d-glucuronide (**34**), respectively.

The molecular formula of peak 28′ (*m/z* 799.19391) was C_33_H_38_O_23_. Its MS/MS fragment ion peaks displayed at *m/z* 623.15869 [M − H − 176], 315.05048 [M − H − 176 − 162 − 146]^−^, 300.02713, 271.02469, and 243.02880 suggested the aglycone of it was isorhamnetin and the substituted sugar moieties included one Glu, one Glc, and one Rha. According to the biosynthesis laws summarized above, peak 28′ was deduced to be isorhamnetin-3-*O*-rutinosyl-7-*O*-β-d-glucuronide (**37**) (Appendix A, Appendix A).

#### 2.3.2. Structural Elucidation of Phenolic Acids

Peaks 1′, 2′, 7′, 12′, and 17′–20′ were identified unequivocally by comparing with reference standards (Appendix A, Figure 7). As what have been mentioned above, the aglycones of phenolic acid glycosides included coumaric acid, caffeic acid, and ferulic acid. It was well known that the characteristic ions of coumaroyl, caffeoyl, and feruloyl were at *m/z* 163.03897 ([coumaroyl − H]^−^), 179.03389 ([caffeoyl − H]^−^), and 193.04954 ([feruloyl − H]^−^), respectively [31]. Then, all of the ions would further generate fragment ion peaks (as shown in Figure 12) by removing 44 Da (–CO_2_), 28 Da (–CO), and 18 Da (–H_2_O), respectively.

Meanwhile, the phenomenon of the neutral loss 120 Da on the basis of [M − H]^−^ were only found in β-d-glucopyranosyl(1→2)-β-d-glucopyranosyl substituted phenolic acid glycosides **7** (peak 7′), **28** (peak 12′), and **29** (peak 17′) (Figure 13 and Appendix A), which could be used to distinguish the type of substituted sugar moieties.

Moreover, comparing the *t*_R_ of compounds **10** (peak 18′) and **29** (peak 17′), α-d-glucopyranosyl-substituted phenolic acid glycoside was found to have the shorter *t*_R_ than that of β-d-glucopyranosyl-substituted ones.

The molecular formula of peak 15′ (*m/z* 487.14313) was C_21_H_28_O_13_. Its MS/MS fragment ion peaks displayed at *m/z* 367.10297, 163.03888, and 145.02829, which was similar to those of peak 12′ (Appendix A, Figure 14). According to the above-mentioned chromatographic retention behavior, we could deduce that peak 15′ was not *p*-hydroxycinnamic acid-9-*O*-α-d-glucopyranosyl(1→2)-β-d-glucopyranoside. As Han et al. reported, the *t*_R_ of *cis*-phenylpropane glycoside was longer than that of *trans* one when they were analysed by HPLC with the acetonitrile-water system [24]. Consequently, peak 15′ was tentatively presumed to be *cis*-*p*-hydroxycinnamate sophorose (**35**).

## 3. Materials and Methods

### 3.1. Materials and Methods for Phytochemistry Research

#### 3.1.1. General Experimental Procedures

UV and IR spectra were recorded on a Varian Cary 50 UV-Vis and Varian 640-IR FT-IR spectrophotometer, respectively. Optical rotations were measured on a Rudolph Autopol^®^ IV automatic polarimeter. NMR spectra were determined on a Bruker 500 MHz NMR spectrometer at 500 MHz for ^1^H and 125 MHz for ^13^C-NMR (internal standard: TMS). Negative-ion mode ESI-Q-Orbitrap MS were obtained on a Thermo UltiMate 3000 UHPLC instrument (Thermo, Waltham, MA, USA).

Column chromatographies (CC) were performed on macroporous resin D101 (Haiguang Chemical Co., Ltd., Tianjin, China), silica gel (48–75 μm, Qingdao Haiyang Chemical Co., Ltd., Qingdao, China), ODS (40–63 μm, YMC Co., Ltd., Tokyo, Japan), and Sephadex LH-20 (Ge Healthcare Bio-Sciences, Uppsala, Sweden). Preparative high performance liquid chromatography (pHPLC) column, Cosmosil 5C_18_-MS-II (20 mm i.d. × 250 mm, Nakalai Tesque, Inc., Tokyo, Japan) were used to separate the constituents.

#### 3.1.2. Plant Material

The fresh aerial parts of *Allium mongolicum* Regel were collected from Alxa League, Inner Mongolia Autonomous Region, China, and identified by Dr. Li Tianxiang (The Hall of TCM Specimens, Tianjin University of TCM, China). The voucher specimen was deposited at the Academy of Traditional Chinese Medicine of Tianjin University of TCM.

#### 3.1.3. Extraction and Isolation

See Appendix A.

#### 3.1.4. Acid Hydrolysis of 1, 3, 4 and 6

Solution of **1**, **3**, **4** and **6** (each 2.0 mg) in 5% aqueous H_2_SO_4_-1,4-dioxane were heated under reflux for 1 h, respectively. After cooling, the reaction mixture was neutralized with Amberlite IRA-400 (OH^−^ form), removed by filtration, subjected to ODS CC (H_2_O), and the H_2_O eluate was reacted with l-cysteine methyl ester hydrochloride in pyridine and *N*,*O*-bis(trimethylsilyl)trifluoroacetamide (BSTFA), successively. Finally, the reaction product was elucidated by GC analysis (GC conditions, column: Agilent Technologies INC Catalog 19,091 J-413 HP-5, 30 m × 0.320 mm (i.d.) capillary column; column temperature: 230 °C; carrier gas: N_2_), and d-glucuronic acid and d-glucose hydrolysates were identified from **1**, **3**, and **6**; d-glucuronic acid, d-glucose, as well as l-rhamnose hydrolysates were detected from **4** by comparing it retention times (*t*_R_: d-glucuronic acid, 23.3 min; d-glucose, 19.6 min; l-rhamnose, 11.4 min) with those of their authentic samples treated in the same way.

#### 3.1.5. Acid Hydrolysis of **2**, **5** and **7**–**10**

The solution of compounds **2**, **5** and **7**–**10** (each 1.5 mg) in 1 M HCl (1.0 mL) was heated under reflux for 3 h. After cooling, the reaction mixture was neutralized with Amberlite IRA-400 (OH^−^ form), then analyzed by HLPC [column, Kaseisorb LC NH_2_-60-5, 4.6 mm i.d. × 250 mm (Tokyo Kasei Co., Ltd., Tokyo, Japan); mobile phase, CH_3_CN-H_2_O (75:25, *v/v*; flow rate, 1.0 mL/min)]. As a result, d-glucose was detected from the aqueous phase of **2**, **5** and **7**–**10** by comparison of its retention time and optical rotation with those of the authentic sample, d-glucose (*t*_R_ 12.5 min, positive), respectively.

### 3.2. Materials and Methods for Bioassay

The activities of compounds **1**–**31** were tested for frequency and height by using tissue perfusion method reported before [30]. Samples in DMSO solution were added after 15 min equilibrate incubation; the final DMSO concentration was 0.1% and final concentration of samples were 50 μM. Mosapride citrate dihydrate (Xi’an Janssen Pharmaceutical Ltd., Xi’an, China), final concentration was 200 μg/mL.

Data were analyzed by SPSS 22.0 software. All values were expressed as mean ± S.D. A *p*-value of 0.05 was considered to indicate statistical significance. One-way analysis of variance (ANOVA) and Tukey’s studentized range test were used for the evaluation of the significant differences between means and post hoc, respectively.

### 3.3. Materials and Methods for Qualitative Analysis

#### 3.3.1. Materials

The isolated 31 compounds including 24 flavonoids, and 7 phenolic acids were used for reference standards. Their purities were > 98%.

HPLC grade Acetonitrile (Thermo-fisher, Waltham, MA, USA), formic acid (Roe Scientific Inc., Newark, NJ, USA), and ultra-pure water prepared with a Milli-Q purification system (Millipore, MA, USA) were used for LC-MS analysis.

#### 3.3.2. Sample Preparation

##### Preparation of Standard Solutions

Standard test solutions of the above-mentioned standard references were prepared in MeOH at a final concentration of 1 μg/mL approximately. All stock solutions were stored at 4 °C in darkness and brought to room temperature before use.

##### Preparation of the Aerial Parts of *A. mongolicum* Extract Test Solutions

*A. mongolicum* extract (AM) was prepared by using the same method as described in “Extraction and Isolation” section. The AM was dissolved with MeOH and filtered with 0.22 µm microporous membrane to get test stock solution at a final concentration of 30 mg/mL, which was stored at 4 °C in darkness and brought to room temperature before use.

#### 3.3.3. UHPLC

A Thermo UltiMate 3000 UHPLC instrument (Thermo, Waltham, MA) equipped with a quaternary pump, an autosampler was used to accomplish the analysis. Samples were separated on a Waters ACQUITY UPLC® HSS C18 (2.1 × 100 mm, 1.8 μm) using a mobile phase composed of H_2_O with 0.1% formic acid (A) and CH_3_CN with 0.1% formic acid (B) in the gradient program: 0–2 min, 9–10% B; 2–5 min, 10–17% B; 5–7 min, 17–20% B; 7–9 min, 20% B; 9–10 min, 20–86% B; 10–14 min, 86–100% B; 14–17 min, 100% B; An equilibration of 3 min was used between successive injections. The flow rate was 0.4 mL/min, and column temperature was 35 °C. An aliquot of 1 μL of each sample was injected for analysis.

#### 3.3.4. ESI-Q-Orbitrap MS and Automatic Components Extraction

For tandem mass spectrometry analysis, a Thermo ESI-Q-Orbitrap MS mass spectrometer was connected to the UltiMate 3000 UHPLC instrument via ESI interface. Ultra-high purity nitrogen (N_2_) was used as the collision gas and the sheath/auxiliary gas. The ESI source parameters were set as follows: ion spray voltage 3.2 kV, capillary temperature 350 °C, ion source heater temperature 300 °C, sheath gas (N_2_) 40 L/h, auxiliary gas (N_2_) 10 L/min, and a normalized collision energy (NCE) of −35 V was used. The Orbitrap analyzer scanned the mass range from *m/z* 150 to 1500 in negative ion mode. Monitoring time was 0–17 min. Detection was obtained by full mass-dd mass mode. The MS data were recorded in both profile and centroid formats. Data recording and processing were performed using the Xcalibur 4.0 software (Thermo Fisher Scientific, Inc., Waltham, MA, USA). The accuracy error threshold was fixed at 5 ppm.

Software-aided, automatic background subtraction and components extraction technique was used to generate a peak list containing all the components profiled from the aerial part of *A.*
*mongolicum*. Sieve v2.2 SP2 (Thermo Fisher Scientific) was used for the automatic components extraction: time range, 1–17 min; BP minimum count, 10,000; BP minimum scans, 5; Background SN, 3; MZ Step, 10; and Frame, >1.

## 4. Conclusions

This paper displayed a study—the first of its kind—focused on the systematic bioactive constituents of the aerial parts of *A. mongolicum* in the gastrointestinal tract. As a result, AM and AMH showed a significant increase in the contraction amplitude of mouse small intestinal muscles, which indicated they might have therapeutic effects on constipation. During this process, we made several achievements:

The first comprehensive phytochemistry investigation was developed for AM by using various spectral and chromatographic methods: six new flavonoid glycosides, mongoflavonosides A_1_ (**1**), A_2_ (**2**), A_3_ (**3**), A_4_ (**4**), B_1_ (**5**), B_2_ (**6**), four new phenolic acid glycosides, mongophenosides A_1_ (**7**), A_2_ (**8**), A_3_ (**9**), B (**10**), as well as 21 known compounds were yielded. They were mainly flavonoids and phenolic acids.

The flavonoids and phenolic acids were clarified for the first time to be the main bioactive constituents of *A. mongolicum* on gastrointestinal tract: flavonoids **3**, **4**, **11**–**15**, **21**–**23**, and **26**, as well as phenolic acids **7**, **29**, and **30** showed significant increase in the height of mouse small intestinal muscle.

Furthermore, a fast analysis method for flavonoids and phenolic acids in the aerial parts of *A. mongolicum* was established for the first time by using LC-MS. According to *t_R_* and *m/z*, 31 compounds (**1**–**31**) were unambiguously identified by comparing to the standard references. Then, on the basis of generalized rules of MS/MS fragmentation pattern, chromatographic behaviors, as well as their biosynthetic laws, five flavonoid glycosides (**32**–**36**) and one phenolic acid glycoside (**37**) were tentatively speculated. Among them, peak **36** was a potential new one. Thus, the first evidence for quality control of *A. mongolicum* has been duly provided.

## Figures and Tables

**Figure 1 molecules-25-00577-f001:**
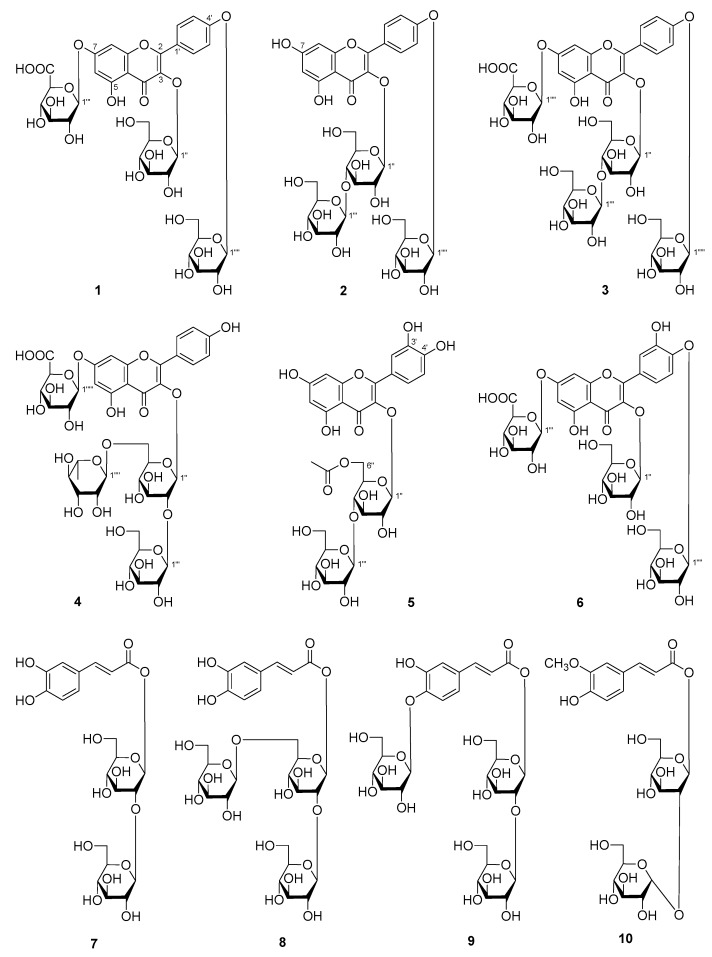
The new compounds obtained from the aerial parts of *A. mongolicum* (**1**–**10**).

**Figure 2 molecules-25-00577-f002:**
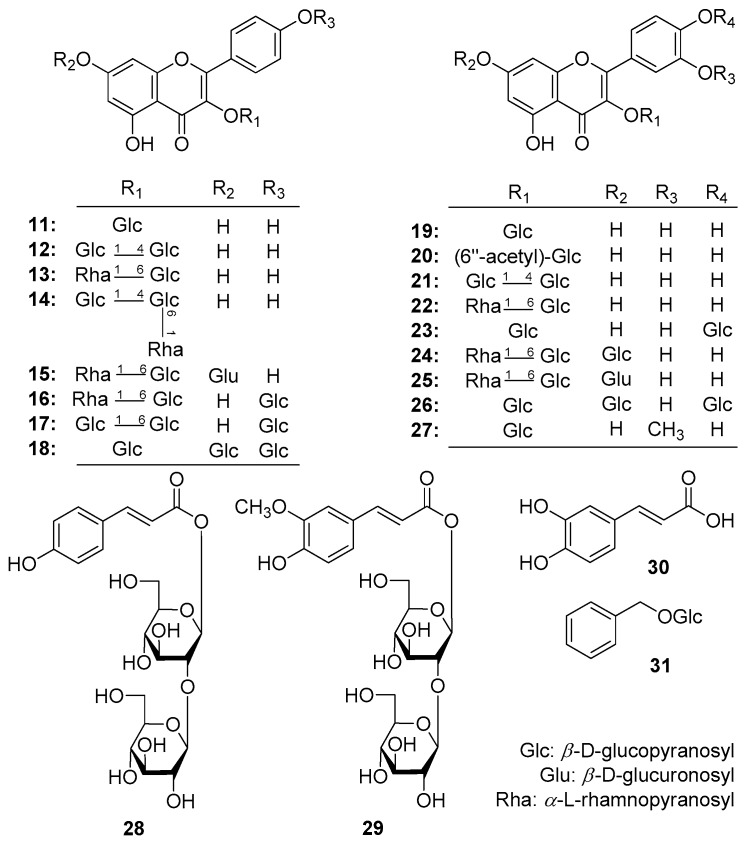
The known compounds obtained from the aerial parts of *A. mongolicum* (**11**–**31**).

**Figure 3 molecules-25-00577-f003:**
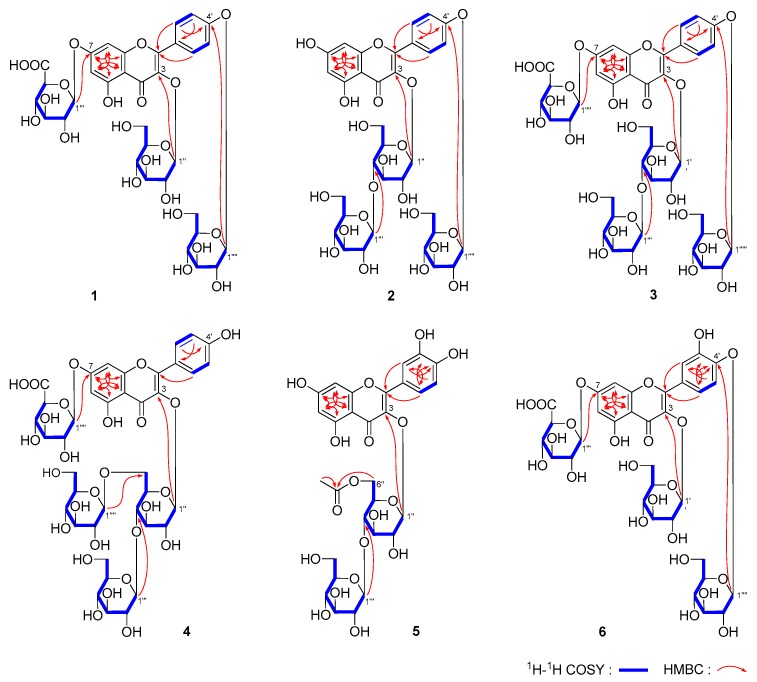
Key ^1^H ^1^H COSY and HMBC correlations of **1**–**6**.

**Figure 4 molecules-25-00577-f004:**
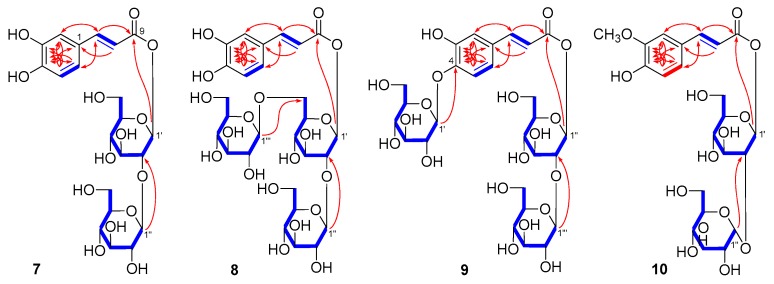
Key ^1^H ^1^H COSY and HMBC correlations of **7**–**10**.

**Figure 5 molecules-25-00577-f005:**
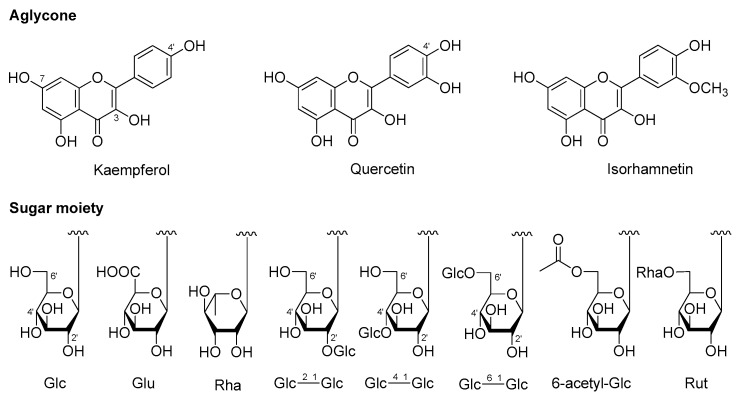
The structure of aglycones and glycosyls of flavonoids from the aerial parts of *A. mongolicum*.

**Figure 6 molecules-25-00577-f006:**
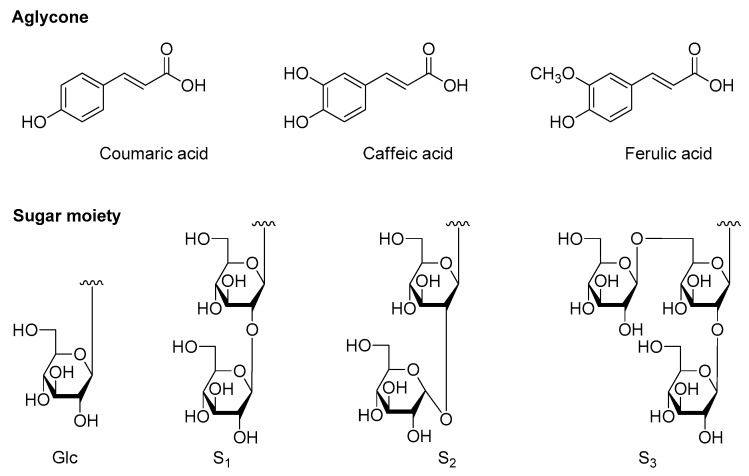
The structure of aglycones and glycosyls of phenolic acids from the aerial parts of *A. mongolicum*.

**Figure 7 molecules-25-00577-f007:**
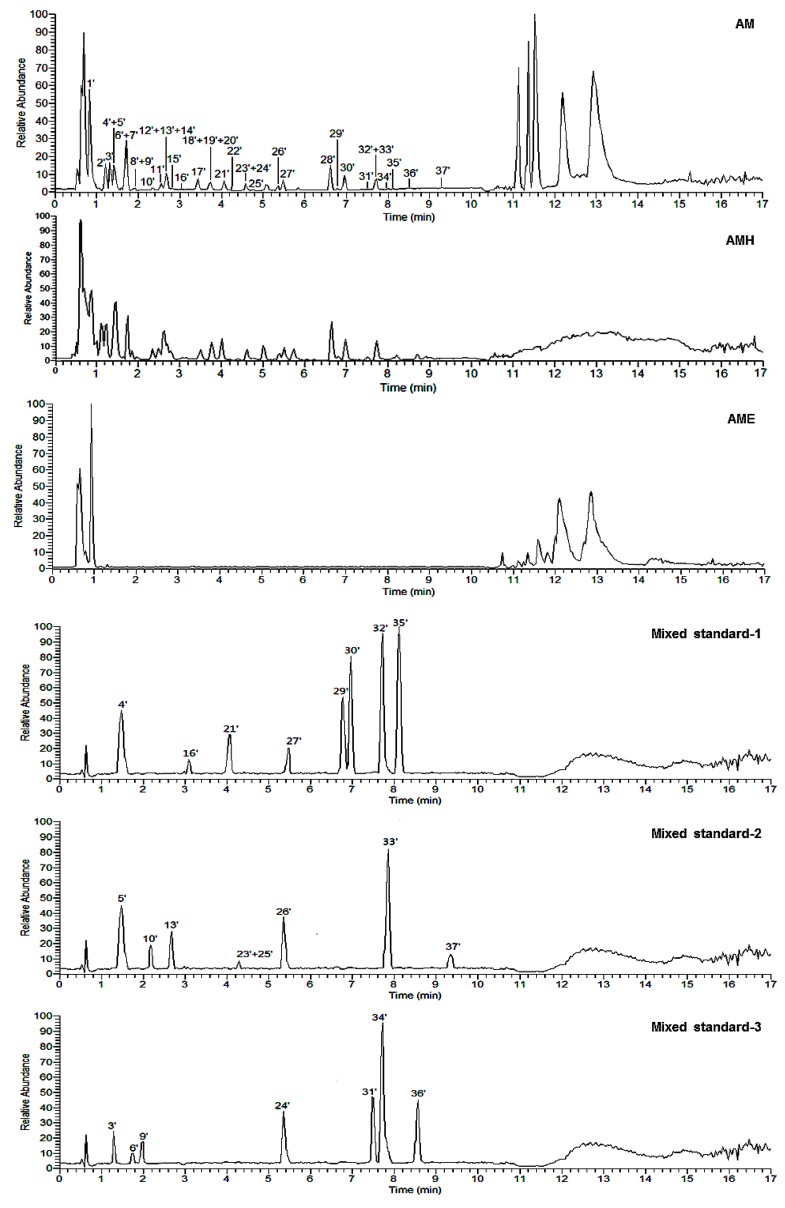
Base peak chromatograms of AM (the extract obtained from fresh the aerial parts of *A. mongolicum* heated reflux with 95% EtOH and 50% EtOH one time each, successively), AMH (H_2_O layer extract), AME (EtOAc layer extract), and mixed standard references.

**Figure 8 molecules-25-00577-f008:**
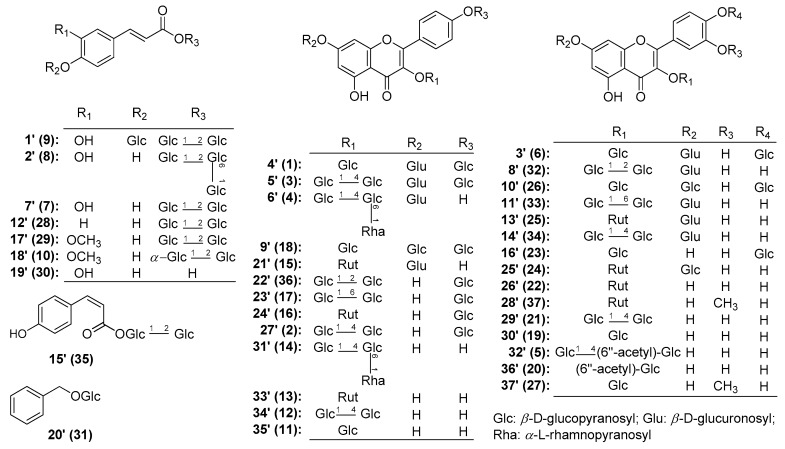
The structures of tentatively presumed compounds from the aerial parts of *A. mongolicum*.

**Figure 9 molecules-25-00577-f009:**
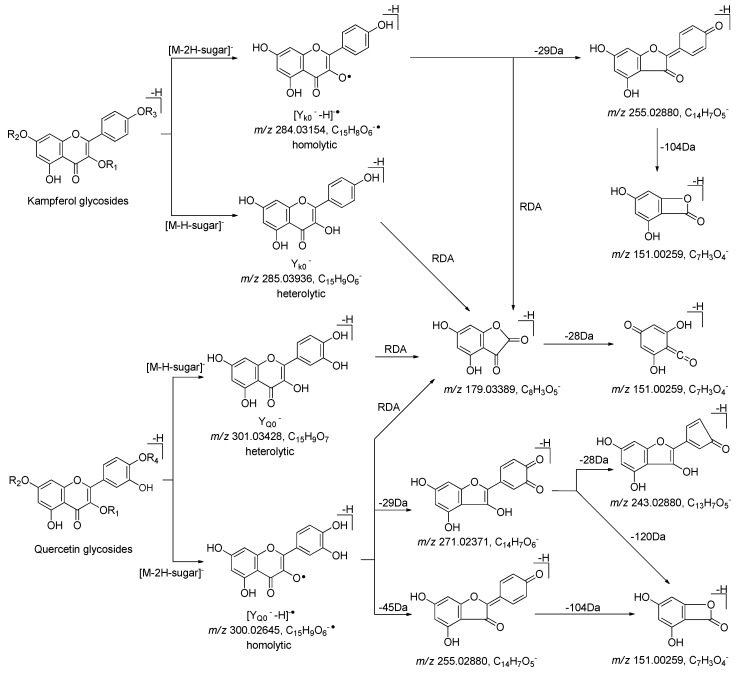
The proposed fragmentation pathways of kaempferol and quercetin glycosides.

**Figure 10 molecules-25-00577-f010:**
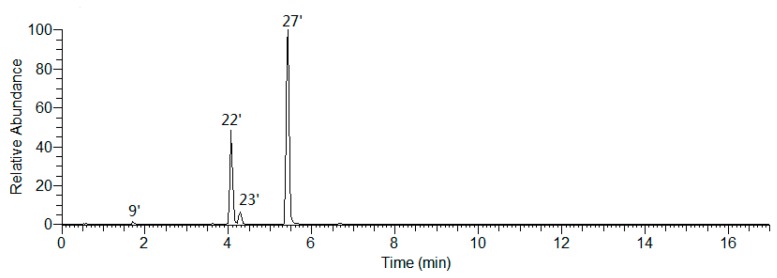
The EIC of the *m/z* 771.19783.

**Figure 11 molecules-25-00577-f011:**
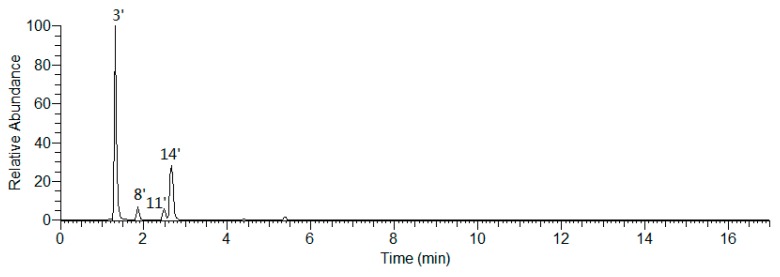
The EIC of the *m/z* 801.17201.

**Figure 12 molecules-25-00577-f012:**
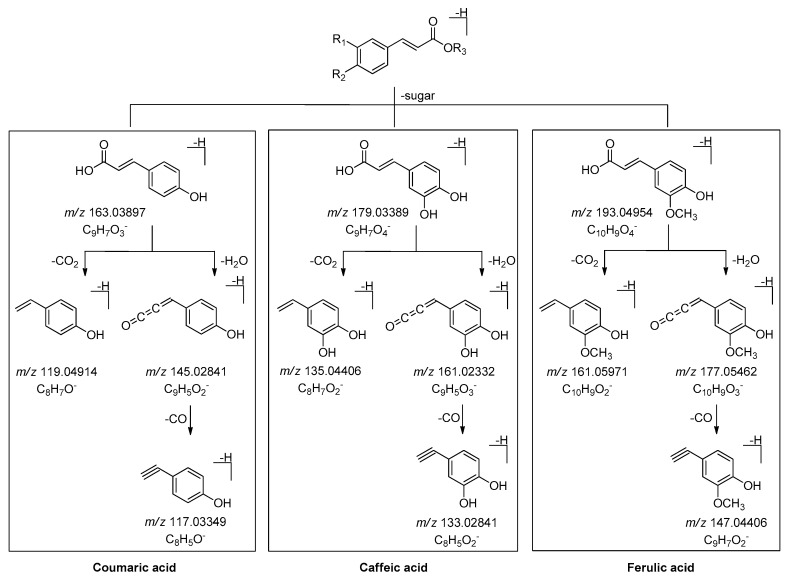
The proposed fragmentation pathways of coumaric acid, caffeic acid, and ferulic acid glycosides.

**Figure 13 molecules-25-00577-f013:**
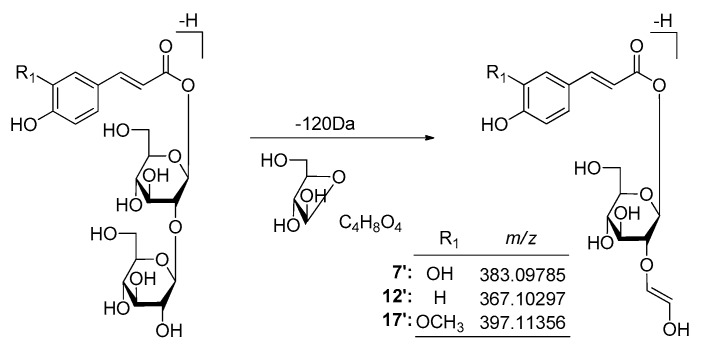
The proposed fragmentation pathways of β-d-glucopyranosyl(1→2)-β-d-glucopyranosyl substituted phenolic acid glycosides.

**Figure 14 molecules-25-00577-f014:**
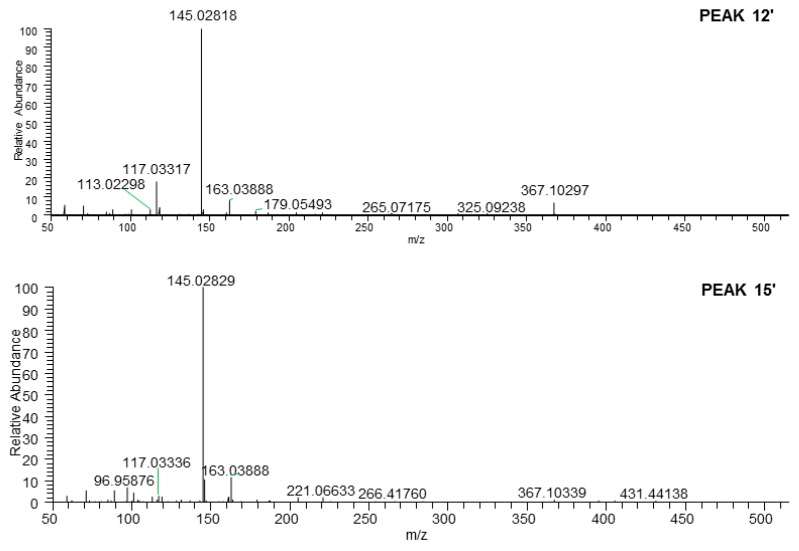
The tandem MS of the [M − H]^−^ ions for peaks 12′ and 15′.

**Table 1 molecules-25-00577-t001:** ^13^C-NMR data for **1**–**6** in DMSO-*d*_6_.

No.	1	2	3	4	5	6	No.	1	2	3	4	5	6
2	156.0	155.5	156.1	157.9	156.3	156.0	*C*OCH_3_					169.6	
3	133.9	133.5	133.9	133.1	132.8	134.0	CO*C*H_3_					19.9	
4	177.7	177.3	177.6	177.3	177.0	177.7	1′′′	99.0	103.0	103.0	103.4	103.2	99.2
5	160.8	161.1	160.7	160.4	161.1	160.7	2′′′	72.6	73.2	73.1	73.0	73.0	72.7
6	99.2	98.9	99.4	99.3	98.8	99.2	3′′′	75.7	76.3	76.3	76.3	76.3	76.1
7	162.5	164.8	162.9	162.3	164.9	162.7	4′′′	71.3	69.9	69.9	69.9	69.9	71.6
8	94.3	93.8	94.4	94.6	93.6	94.3	5′′′	75.0	76.7	76.7	76.7	76.7	74.3
9	156.2	156.4	156.0	156.2	156.5	156.0	6′′′	170.5	60.9	60.9	60.9	61.0	171.4
10	105.8	103.8	105.6	105.6	103.4	105.6	1′′′′	99.8	99.9	99.3	101.0		101.3
1′	123.5	123.6	123.4	120.7	120.8	124.2	2′′′′	73.1	73.1	72.8	70.4		73.2
2′	130.6	130.5	130.6	130.9	116.0	116.5	3′′′′	76.4	76.4	76.2	70.9		75.7
3′	115.7	115.7	115.7	115.0	144.8	146.0	4′′′′	69.5	69.5	71.7	71.6		69.6
4′	159.2	159.1	159.3	159.9	148.5	147.5	5′′′′′	77.0	77.0	73.9	68.1		77.1
5′	115.7	115.7	115.7	115.0	115.0	115.3	6′′′′′	60.5	60.5	171.4	17.5		60.6
6′	130.6	130.5	130.6	130.9	121.3	121.0	1′′′′′			99.8	98.7		
1′′	100.6	100.5	100.4	100.8	100.5	100.5	2′′′′′			73.1	72.8		
2′′	74.1	73.8	73.1	73.7	73.5	74.0	3′′′′′			76.4	76.2		
3′′	76.3	74.6	74.6	74.6	74.5	76.4	4′′′′′			69.5	71.6		
4′′	69.8	80.0	80.0	80.8	80.0	69.9	5′′′′′			77.0	73.7		
5′′	77.5	75.3	75.4	73.7	71.9	77.6	6′′′′′			60.5	171.7		
6′′	60.8	60.1	60.0	67.0	62.1	60.8							

**Table 2 molecules-25-00577-t002:** ^13^C-NMR data for **7**–**10**.

No.	7 *^a^*	8 *^a^*	9 *^a^*	10 *^b^*	No.	7 *^a^*	8 *^a^*	9 *^a^*	10 *^b^*
1	125.4	125.2	128.5	127.5	5′	77.4	75.9	77.1	78.0
2	114.6	114.6	115.0	111.7	6′	60.3	67.7	60.6	62.8
3	145.5	145.6	146.7	149.5	1′′	104.4	104.4	92.4	94.0
4	148.6	149.0	147.5	151.0	2′′	74.4	74.4	81.6	73.9
5	115.6	115.6	115.9	116.6	3′′	76.0	76.1	75.6	74.9
6	121.6	121.7	120.9	124.2	4′′	69.3	69.3	69.0	71.8
7	146.2	146.3	145.5	146.9	5′′	76.6	76.7	77.5	73.0
8	113.3	113.1	115.5	115.1	6′′	60.3	60.3	60.3	62.9
9	164.9	164.9	164.7	169.8	1′′′		103.0	104.5	
3-OCH_3_				56.5	2′′′		73.4	74.5	
1′	92.3	92.3	101.4	98.2	3′′′		76.6	76.0	
2′	81.5	81.4	73.1	76.3	4′′′		69.9	69.3	
3′	75.6	75.5	75.7	78.1	5′′′		76.8	76.6	
4′	69.0	68.8	69.7	71.9	6′′′		60.9	60.3	

Determined in *^a^* DMSO-*d*_6_, *^b^* CD_3_OD.

**Table 3 molecules-25-00577-t003:** Inhibitory effects of compounds **1**–**31** on motility of mouse isolated intestine tissue.

Compd.	Intestine Motility (%)		Compd.	Intestine Motility (%)	
	Relative Height	Relative Frequency		Relative Height	Relative Frequency
N	100.0 ± 4.9	100.0 ± 1.2	**16**	116.1 ± 10.4	81.3 ± 11.1
P	190.8 ± 19.2 **	85.6 ± 2.6	**17**	108.1 ± 6.1	97.3 ± 2.3
**1**	112.3 ± 2.3	99.4 ± 1.2	**18**	116.6 ± 4.4	106.6 ± 4.5
**2**	105.1 ± 19.5	98.8 ± 2.9	**19**	107.7 ± 3.1	99.3 ± 3.5
**3**	148.9 ± 4.5 **	100.8 ± 0.9	**20**	107.8 ± 32.7	99.4 ± 0.6
**4**	170.0 ± 6.4 *	98.6 ± 2.1	**21**	121.9 ± 6.6 *	95.2 ± 4.4
**5**	99.7 ± 25.0	101.6 ± 7.7	**22**	125.2 ± 8.1 *	97.4 ± 4.0
**6**	107.9 ± 18.9	96.7 ± 3.1	**23**	142.2 ± 11.2 *	101.1 ± 3.7
**7**	157.4 ± 20.8 *	94.9 ± 2.7	**24**	117.9 ± 12.7	98.3 ± 2.8
**8**	150.5 ± 25.9	95.6 ± 2.8	**25**	104.9 ± 7.7	96.3 ± 1.9
**9**	149.1 ± 36.2	94.6 ± 3.8	**26**	137.4 ± 2.4 **	96.2 ± 2.8
**10**	121.1 ± 21.6	98.7 ± 3.0	**27**	105.6 ± 32.3	91.8 ± 3.8
**11**	123.2 ± 6.8 *	99.1 ± 7.1	**28**	136.8 ± 12.4	97.1 ± 2.0
**12**	144.2 ± 14.3 *	95.7 ± 1.3	**29**	148.1 ± 6.8 *	105.7 ± 4.1
**13**	151.5 ± 17.1 *	98.8 ± 4.4	**30**	125.0 ± 1.6 **	98.8 ± 2.3
**14**	120.4 ± 9.2 *	98.4 ± 1.7	**31**	97.8 ± 1.9	98.8 ± 1.2
**15**	143.7 ± 1.3 **	100.6 ± 3.1			

Values are the means ± standard error of measurement, significantly different from the control group, * *p* < 0.05, ** *p* < 0.01, n = 6. Normal (N): isolated intestine tissue; Positive control (P): Mosapride citrate dihydrate, final concentration was 200 μg/mL. Compounds **1**–**31**: final concentration was 50 μM. Frequency, and height of normal group was set as 100%, relative values of them were calculated as: (sample/normal) × 100%.

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
