# Peer review of "Phytochemistry and Comprehensive Chemical Profiling Study of Flavonoids and Phenolic Acids in the Aerial Parts of Allium Mongolicum Regel and Their Intestinal Motility Evaluation"

_molecules, 2020, doi:10.3390/molecules25030577_

Round 1

Reviewer 1 Report

The study by Dong et al. presents the isolation and identification by MS, NMR and FTIR techniques of flavonoids and phenolic acid from A. Mongolicum, indicating that these are the main constituents of A. mongolicum. A second part of the work deals with the study on motility of mouse isolated intestine.

Althaugh extensive work was performed by the authors, the manuscript has to be considerable improved:

In my opinion, the isolation and identification of A. Mongolicum described by the 14 figures from the total of 15 already exeed the length of a journal, therefore I recommend to publish the bioassay part as another, separate study. The title:

Please correct Mongolicum instead Mongolicm

The Abstract section:

Trivial affirmations such as:

“As an important edible medicinal plant, Allium mongolicum Regel is not only with unique flavor and high nutritional value, but also has various special effects such as stimulating the appetite”, should be avoided in the abstract section. Moreover, the mentioning of compounds in the form of numbers such as „ flavonoids 3, 4, 1115, 2123, and 26, as well as phenolic acids, 7, 29, and 30” should be avoided. The abstract section should mention the aim of the work, the used methods, the gained results and clearly the novelty of the results.

Introduction section:

As the manuscript is focused on isolation and identification of phytochemicals from A. Mongolicum, healthy and pharmacological properties mentioned in the introduction part should be shortened and instead, a more detailed literature analysis on phytochemicals in Allium species should be presented, with emphasis on the flavonoid and phenolic acid compounds.

Language and spelling should be considerable improved.

Reviewer 2 Report

The authors assembled a diligent research for the identification of a specific family of metabolites in Alium Mongolicum and to evaluate their effect on intestinal motility. The study and the results are well described and in general I think that the manuscript could deserve publication on the journal after the following revisions: 

1) Section 2. "The fresh A. mongolicum (17.8 kg) was successively heated reflux with 95% EtOH and 50% EtOH one time each to obtain A. mongolicum extract (AM, 515.0 g). Then 470.1 g of it was partitioned with EtOAc/H2O (1:1, v/v) to yield EtOAc layer extract (AME, 64.9 g) and H2O layer extract (AMH, 381.0 67 g).": Do the masses refer to the dry extracts or to the solutions? What was the volume of the EtOAc/H2O mixture used for the second partition?

2) Table 3: Where the significances obtained after correcting for multiple testing? If so, which approach was followed?

3 - Major) Given the multivariate nature of the data set, why no multivariate approach to evaluate the effect of the different metabolites on the gut mobility was followed? Univariate analysis only provides a partial picture when the overall metabolic variation is involved. 

Reviewer 3 Report

First of all it is really worth-appreciating that the results reported in the manuscript effect from huge amount of really hard work. This is particularly visible in case of the preparative chromatographic separation of the investigated compounds. 

On the other hand, the manuscript suffers from a number of serious issues which have to be address prior the final acceptance for publication. These issues are as follows:

The whole manuscript requires a detailed and professional language editing

Subsection 2.1. A long list of spectroscopic characteristics of subsequent isolated compounds is hard to read. I would suggest a change in the way data is presented. Some of them are already placed in tables so in the text it is better to place only the most important, characteristic values. This would make the manuscript easier for readers to read and understand.

Fig 7: What type of detector was used to get the results presented? Are those results obtained after MS analysis, or did authors used some other, e.g. UV detector after the column?

Line 484: Why the authors refer to indica.

Supplementary material:

Extraction and Isolation: What part of the plant did authors use as the raw material for the extraction? Was it the whole plant (roots, stems, leaves, flowers) or only the selected parts?

How long did the heating and refluxing take? Why was it performed twice and at various concentrations of ethanol? What was the duration of each of these heatings?

Chromatography conditions, except of UHPLC needs better clarification.

How high was the flow rate of eluents in each separation run?

What the duration of each separation process?

Did the eluents change (linearly or in steps) or was it isocratic process?

What was the temperature of the column?

What type of detector/detectors was/were used for fractions monitoring?

In a conclusion I strongly recommend a major revision of the manuscript but also strongly encourage the Authors to improve their overall original and interesting contribution.

Round 2

Reviewer 2 Report

I am satisfied with the authors' replies to the comments and the manuscript can now be accepted for publication.

Author Response

Thank you for your kindness.

Reviewer 3 Report

The issues were addressed properly, and the current version of the manuscript is an improvement compared to the previous one. I recommend its publication after final spell check.

Author Response

Thank you for your suggestion. We have checked and corrected the mistake spell in the MS again.